# High-Precision Single Building Model Reconstruction Based on the Registration between OSM and DSM from Satellite Stereos

**Yong He, Wenting Liao, Hao Hong and Xu Huang \***

School of Geospatial Engineering and Science, Sun Yat-Sen University, Zhuhai 519082, China
\* Correspondence: huangx358@mail.sysu.edu.cn; Tel.: +86-1597-220-7433

**Abstract:** For large-scale 3D building reconstruction, there have been several approaches to utilizing multi-view satellite imagery to produce a digital surface model (DSM) for height information and extracting building footprints for contour information. However, limited by satellite resolutions and viewing angles, the corresponding DSM and building footprints are sometimes of a low accuracy, thus generating low-accuracy building models. Though some recent studies have added GIS data to refine the contour of the building footprints, the registration errors between the GIS data and satellite images are not considered. Since OpenStreetMap (OSM) provides a high level of precision and complete building polygons in most cities worldwide, this paper proposes an automatic single building reconstruction method that utilizes a DSM from high-resolution satellite stereos, as well as building footprints from OSM. The core algorithm accurately registers the building polygons from OSM with the rasterized height information from the DSM. To achieve this goal, this paper proposes a two-step "coarse-to-fine registration" algorithm, with both steps being formulated into the optimization of energy functions. The coarse registration is optimized by separately moving the OSM polygons at fixed steps with the constraints of a boundary gradient, an interior elevation mean, and variance. Given the initial solution of the coarse registration, the fine registration is optimized by a genetic algorithm to compute the accurate translations and rotations between the DSM and OSM. Experiments performed in the Beijing/Shanghai region show that the proposed method can significantly improve the *IoU* (intersection over union) of the registration results by 69.8%/26.2%, the precision by 41.0%/15.5%, the recall by 41.0%/16.0%, and the F1-score by 42.7%/15.8%. For the registration, the method can reduce the translation errors by 4.656 m/2.815 m, as well as the rotation errors by $0.538°/0.228°$, which indicates its great potential in smart 3D applications.

**Keywords:** OSM; DSM; single building model; coarse registration; fine registration; genetic algorithm

## 1. Introduction

CityGML of The Open Geospatial Consortium (OGC) categorizes building models into five levels of detail (LOD), which are distinguished by the geometric and semantic complexity of 3D city models [1,2]. When compared with aerial LiDAR data, satellite images are an economic solution for large-scale building reconstruction, due to their high-frequency worldwide imaging mode and increasing spatial resolutions [3,4]. LOD reconstruction from satellite imagery has a great potential in large-scale city modeling, which has various applications in urban planning [5–8], environmental monitoring [9,10], virtual city tours [11], and national defense military [12].

For these large-scale reconstruction applications, several approaches have been proposed towards 3D reconstruction with multi-view satellite imagery. In traditional building reconstruction approaches, the first step is to produce a digital surface model (DSM) by dense image matching (DIM) [12,13]. Since traditional DIM, especially the DIM with two views, often meets edge-fattening issues [14], the reconstruction accuracy around the building contour is not good, which influences the single building reconstruction accuracy, as well as the reconstruction completeness. For more accurate building reconstruction, most

methods extract the building contours from satellite images with deep learning techniques, which can be generally categorized into two major types: building segmentation and contour line detection approaches. The segmentation-based approaches compute image segmentation with a deep learning network to extract building masks in a raster format, and then obtain the building contours via polygonization [4,5,15–21]. The contour line detection approaches first detect the building edges and then group the line segments to generate complete building polygons [22–24], where deep learning techniques are used fully on the extractions of deep edge features, as well as on the optimizations of the line grouping. After extracting the building contours, single building models are reconstructed by assigning the height values from the DSM to them. These deep-learning-based methods can establish 3D building models automatically, while they still meet the challenging issue of the model transferability. In addition to using deep learning and DSMs, Duan and Lafarge [3] applied a geometric algorithm that decomposes images into atomic convex polygons to both stereo images, then retrieved the semantic class and the elevation of two matching polygonal partitions simultaneously to reconstruct the city model. However, such a method could not capture small geometries. All of the approaches mentioned above extract the building contours from satellite imagery. However, the limited quality of satellite images makes it difficult to extract the building contours precisely. The contours may be partially missing due to shadows, occlusions, complex building structures, the differences between the training dataset and the testing dataset (e.g., seasonal changes and spatial resolution differences), and so on [25]. It is also hard to capture small buildings, typically houses in residential areas, due to the limitation of the spatial resolution of satellites. Since many map platforms provide accurate building polygons globally, it is efficient to use the GIS data from these map platforms for robust building contour extraction. Several recent studies utilized GIS data to refine the results of building extraction. Li et al. [26] focused on adding GIS data into the validation dataset for training a building segmentation network and finally improved the F1-score of the network. Gui et al. [27] used OpenStreetMap (OSM) and graph cut labeling to refine the orientation of 2D building polygons computed by a deep-learning-based detector. Esch et al. [28] generated a building mask based on the joint processing of OSM and SAR (synthetic-aperture radar) images. Due to the large complementarity in the physical characterization between the SAR and the optical data [29], their joint use for building reconstruction is a promising direction. However, due to positional errors, there are registration errors between GIS data and satellite images, which are not considered in the approaches above.

Since OpenStreetMap provides high-precision and complete building polygons for most cities worldwide [30,31], utilizing the building contours from OSM directly is an alternative way of performing large-scale building contour extraction. Inspired by the availability of OSM, this paper proposes an automatic single building reconstruction method that utilizes high-precision, DIM-derived DSM from high-resolution satellite stereos, as well as the building footprints from OpenStreetMap. The core algorithm is to accurately register the polygon information from OSM and the rasterized height information of the DSM. To achieve this goal, this paper proposes a two-step "coarse-to-fine registration" algorithm to eliminate the registration errors between OSM and the DSM. Considering the complex systematic errors in the registration, each building is processed separately. In the "coarse registration" step, we first sample several boundary points and interior points from each building, and then compute the optimal translation parameters based on a score composed of the boundary constraints and interior constraints (e.g., the gradient of the boundary points, the elevation, and its variance of the interior points). In the "fine registration" step, we formulate the registration problem as the optimization of an energy function, and compute the optimal translation and rotation parameters based on a genetic algorithm [32–34]. Our main contribution is to innovatively propose a low time-cost, scalable, and high-accuracy single building reconstruction method, based on a DSM and OSM.

The remainder of this paper is organized as follows: Section 2 presents the details of the coarse registration and fine registration algorithms, respectively. The experimental results, discussion, and conclusion are presented in Sections 3–5, respectively.

## 2. Methodology

### 2.1. Workflow

This paper proposes an efficient approach for single building model reconstruction with OSM constraints. The core idea is to optimize the registration between the rasterized DSM and the building footprints from OSM. The overall framework of the proposed method is demonstrated in Figure 1, which formulates the registration into two-step optimizations, first with coarse registration, and then with fine registration.

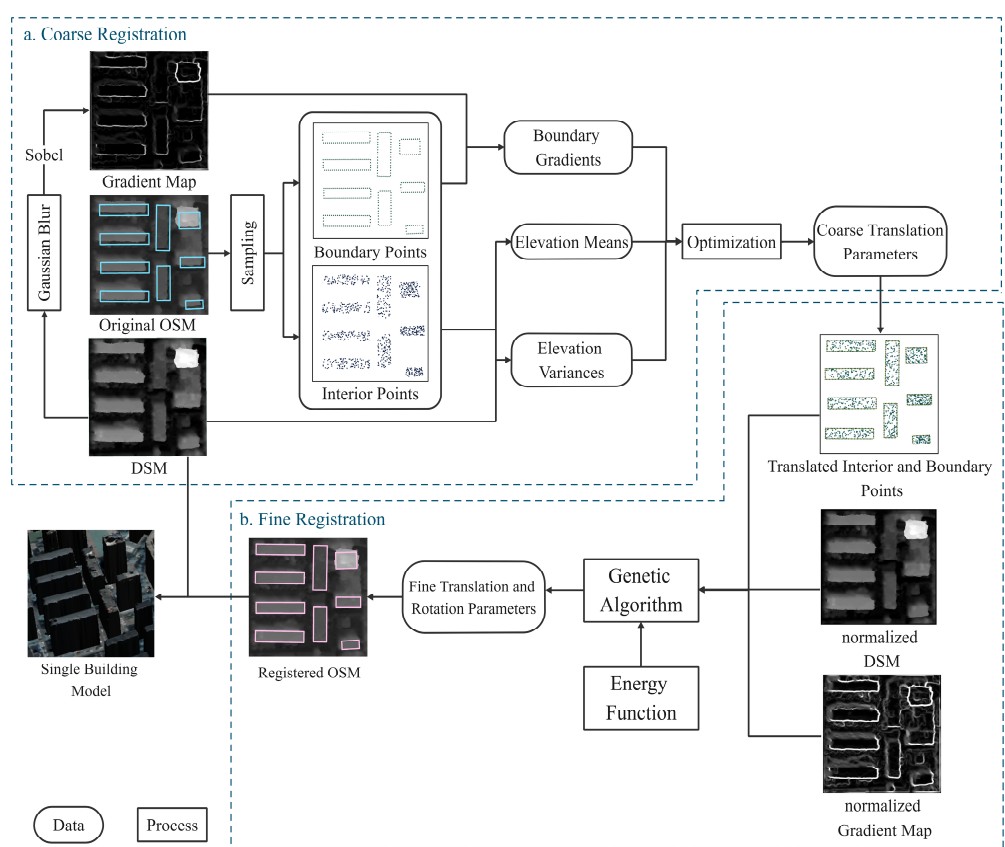

**Figure 1.** An overview of our proposed method. In (**a**) "coarse registration" step, taking DSM, OSM, and the gradient map as input, this paper computed coarse translation parameters by optimization. In (**b**) "fine registration" step, the optimal translation and rotation parameters are computed based on an energy function and genetic algorithm. Single building model was reconstructed by clipping DSM with registered OSM.

The inputs of the coarse registration are OSM, the DSM, and its gradient map, using a Gaussian blur and the Sobel operator, where OSM building points are sampled into a series of boundary points and interior points for efficiency purposes. Since the satellite stereo-derived DSM has little rotation differences to OSM, the coarse registration only considers the translations between the DSM and OSM, and defines an optimization function as a summation of three scores: the boundary gradients from the boundary points, the elevation means, and the variances from the interior points. A brute solution with fixed moving paces is utilized to iteratively compute the optimal coarse translation parameters. In the fine registration step, both the boundary and interior points are translated by the coarse registration results, which will provide a good initial solution to the nonlinear optimization of the fine registration. To acquire high-accuracy registration results, this paper adopts

the rotation and translation parameters as independent arguments, and formulates the fine registration into the optimization of an energy function, whose optimal solution is computed by a genetic algorithm [35,36]. The energy function of the fine registration is implicit with a summation of the three scores of the boundary gradients, elevation means, and variances. Both the DSM and its corresponding gradient map are normalized for the purpose of limiting the weights of the three scores in the range of 0 to 1, which is beneficial for finding the optimal weight parameters in the energy function during the experiments. After the two-step "coarse-to-fine registration" algorithm, a single building model can be built by clipping the DSM with the registered OSM.

*2.2. Coarse Registration*

Coarse registration separately moves the OSM polygons at fixed steps to bind the buildings in the DSM. Assuming that the surrounding buildings with short space distances (e.g., less than 5 m) have similar registration errors, this paper therefore groups the OSM polygons with the short space distances, and assigns the same registration parameters (including the coarse and fine registrations) to the polygons within the same group during the registration process.

The core idea of the coarse registration is to move the OSM polygons until they find the location in the DSM with the most significant elevation gradients. However, it is possible that the object with the highest elevation gradients may not be buildings, e.g., the grounds between the adjacent buildings. To further improve the registration accuracy, this paper additionally considers the elevations within the OSM polygons, so that these non-building objects can be excluded. Since most buildings have high-elevation roofs and vertical facades, this paper considers that the correct building location is where the boundary gradient $\overline{g_m}$ is large, the interior elevation mean $\overline{e_m}$ is large, and the interior elevation variance $\overline{e_v}$ is small. The computation of the three parameters ($\overline{g_m}, \overline{e_m}, \overline{e_v}$) in each building group and the optimization in the coarse registration are shown below.

In general, the inputs of the coarse registration are OSM, the DSM, and its gradient map. For the purpose of efficient computation, this paper first samples a series of boundary points and interior points from OSM before the optimization of the coarse registration. The boundary points are evenly generated along the OSM boundaries at fixed intervals of four times the GSD (ground sample distance), as shown in Figure 2a. The interior points are randomly generated inside the building polygons of OSM, where a polygon could contain 100 points at most, and the minimum allowed distance between these random points is 2 times the GSD, as shown in Figure 2b. Figure 2c,d show the DSM and its corresponding gradient map, respectively.

Considering the noises in the DSM, a low-pass filtering with a $5 \times 5$ gaussian kernel is needed before the coarse registration. Since the building boundaries often cause significant elevation gradients, this paper therefore computes the gradient map $I_g$ using the Sobel gradient operator [37–39] on the blurred DSM $I_S$.

Since the satellite images could be geo-coded from the corresponding RFM (rational function model) parameters, the satellite stereo-derived DSM has little rotation differences to OSM. Thus, the coarse registration only considers the translations between the DSM and OSM for computation purposes. To achieve the optimal coarse translation parameters, each building group is moved in $x$ and $y$ directions, respectively, where the moving step $s_c$ is six times the GSD. Since the maximum translation error in the horizontal or vertical directions is determined with the satellite position error, which varies a lot with different satellite imagery, the maximum moving distance $\delta_c$ of $x$ and $y$ is user-defined. For example, the ground location accuracy without the GCPs of the World View-3 and Pléiades imagery is 3.5 m and 10 m, respectively. The moving step is set as six times the GSD for two reasons: (1) high-resolution satellite images have sub-meter resolutions (e.g., GSD: 0.5 m), so that the moving step (e.g., 3 m) is able to assure the accuracy in the coarse registration when the maximum translation error between OSM and the DSM reaches 10 m; and (2) if the moving step is smaller than six times the GSD, the computation efficiency of the coarse registration

will be reduced, causing a higher time cost. Thus, the moving step is set as 6 times the GSD in the paper, considering the registration accuracy and time cost.

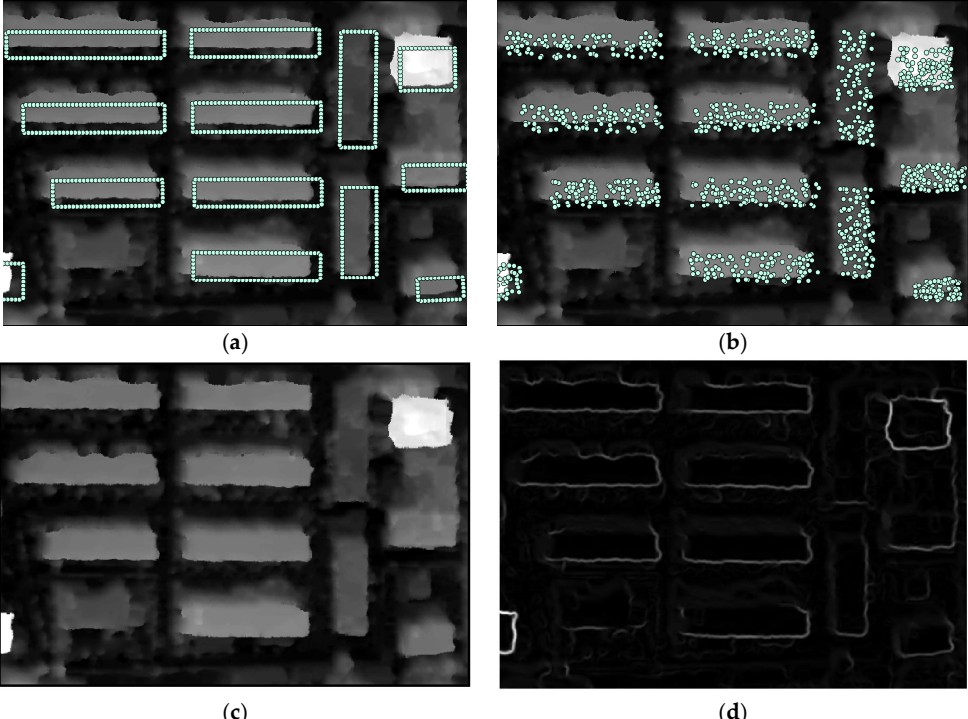

**Figure 2.** The inputs of the coarse registration in an experimental area in Beijing, including (**a**) discrete boundary points of OSM, (**b**) discrete interior points of OSM, (**c**) DSM and (**d**) the corresponding gradient map.

During moving, the boundary and interior points of a building group are translated according to the moving steps in the $x$ and $y$ directions, where the boundary points are available for the elevation gradient computations and the interior points are used to compute the elevation means and variances of the building groups. In general, the best translation is the solution with the maximum boundary gradients, the maximum interior elevations, and the minimum interior variances. The mean gradient $\overline{g_m}(x, y)$ of the boundary points at step $(x, y)$ for a building group is computed on $I_g$. On the other hand, the elevation means and variances are formulated as the weighted averages of all the buildings in the same group. Since the areas of the buildings in the same group are different, this paper assumes that the larger-area buildings should contribute more in the coarse registration, and defines these contributions as weights in the computation of the elevation means $\overline{e_m}$ and elevation variances $\overline{e_v}$ for a certain building group. Each building $b$ in a building group has its own elevation mean $e_m^b(x, y)$ and variance $e_v^b(x, y)$ of the interior points at step $(x, y)$ on $I_S$, and the weight of $b$ (i.e., $w_b$) is determined by the proportion of its area $a_b$ in the group, which is calculated according to Equation (1), where $m$ is the total number of buildings in the group. Thus, the area-weighted averages $\overline{e_m}(x, y)$ and $\overline{e_v}(x, y)$ for the building group at step $(x, y)$ are computed according to Equation (2).

$$w_b = \frac{a_b}{\sum_{b=1}^{m} a_b} \tag{1}$$

$$\overline{e_m}(x, y) = \sum_{b=1}^{m} w_b \times e_m^b(x, y), \quad \overline{e_v}(x, y) = \sum_{b=1}^{m} w_b \times e_v^b(x, y) \tag{2}$$

After moving, each step $(x, y)$ has three corresponding scores: $\overline{g_m}(x, y)$, $\overline{e_m}(x, y)$, and $\overline{e_v}(x, y)$. To balance the contributions of these three scores in the coarse registration, the

minimum and maximum values of each score among all the steps are utilized to normalize $\overline{g_m}(x,y)$, $\overline{e_m}(x,y)$, and $\overline{e_v}(x,y)$, and the normalized scores can be described as $g'_m(x,y)$, $e'_m(x,y)$, and $e'_v(x,y)$. The optimization function consisting of the three normalized scores is then defined to compute the optimal translation parameters $(x,y)$ for each building group, which is shown in Equation (3), where $w_1, w_2$, and $w_3$ are the weights of $g'_m, e'_m$, and $e'_v$ respectively, and $\delta_c$ is the maximum moving distance of $x$ and $y$. As mentioned in this section before, this paper assumes that the optimal building location should be the one with high boundary gradients, a high interior elevation mean, and a low elevation variance; thus, the sign of the variance part in Equation (3) is different from the other two parts. The steps $(x,y)$ with the maximum $S(x,y)$ are the optimal translation parameters for each building group. The details of selecting the optimal weights $w_i$, $i = 1, 2, 3$ are shown in Section 3. The effect of the coarse registration is shown in Figure 3, where Figure 3a,b displays the original OSM and the translated OSM after the coarse registration, respectively.

$$\max_{\substack{-\delta_c \le x \le \delta_c \\ -\delta_c \le y \le \delta_c}} S(x,y) = \max_{\substack{-\delta_c \le x \le \delta_c \\ -\delta_c \le y \le \delta_c}} (w_1 \times g'_m(x,y) + w_2 \times e'_m(x,y) - w_3 \times e'_v(x,y)) \quad (3)$$

where, $S$ is the energy function of the coarse registration; $x, y$ are the translations in the ground space; and $\delta_c$ is the pre-defined range of the translations.

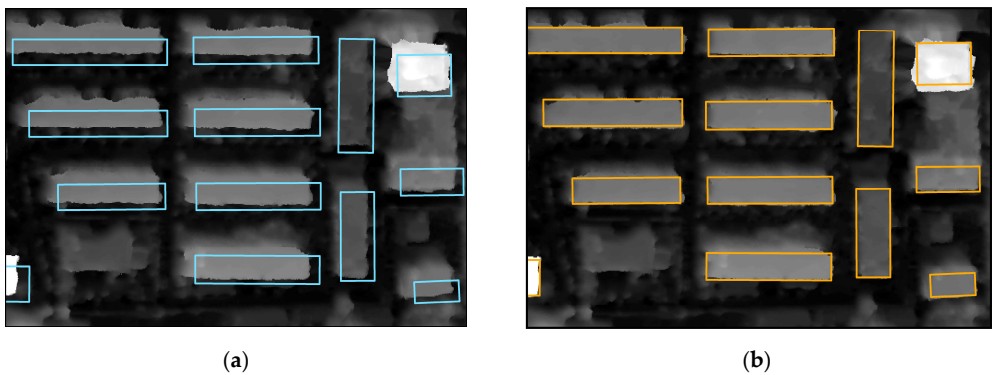

|  (a)  |  (b)  |

**Figure 3.** The effect of the coarse registration. The blue polygons in (**a**) are the original OSM and the orange polygons in (**b**) are the translated OSM after the coarse registration.

*2.3. Fine Registration*

2.3.1. Formulation

Based on the good initial solution provided by the coarse registration, the fine registration is developed to compute the optimal fine translation parameters $x, y$ and the fine rotation parameter $\varphi$ for each building group with a genetic algorithm. Similar to the coarse registration, the core idea of the fine registration is to iteratively move and rotate the OSM polygons to an appropriate location in the DSM, with the constraints of the elevation gradients $g_m$, interior elevation means $e_m$, and elevation variances $e_v$. In general, the fine registration is formulated into the optimization of an energy function, with the independent variables being the translations and rotations between the DSM and OSM buildings, as shown in Equation (4). The basic assumption of the energy function is that the optimal building location should be the one with the high boundary gradients, high interior elevation means, and low elevation variances. Thus, the above three constraints are formulated as the three scores in the energy function. For the purpose of balancing the contributions of these three scores, the DSM and its corresponding gradient map are pre-processed to get the normalized digital height model $I_H^n$ and the normalized gradient map $I_g^n$. Therefore, the inputs of the energy function of the fine registration include the mean gradients $g''_m(x,y,\varphi)$ of the boundary points on $I_g^n$, the elevation means $e''_m(x,y,\varphi)$ of the interior points on $I_H^n$, and the elevation variances $e''_v(x,y,\varphi)$ of the interior points on $I_H^n$

under the transformation parameters $(x, y, \varphi)$, where $x, y$ are the translation parameters and $\varphi$ is the rotation parameter. Since the local ground of the building groups in both OSM and the DSM is approximate to a plane, the single rotation parameter $\varphi$ is enough to correct the rotations between the OSM and DSM buildings. $w_1'$, $w_2'$, and $w_3'$ are defined as the weights of the three scores, which influence the fine registration results. The optimal weights $w_i'$, $i = 1, 2, 3$ will be discussed in Section 3.

$$\min E(x, y, \varphi) = \min\left[-\left(w_1' \times g_m''(x, y, \varphi) + w_2' \times e_m''(x, y, \varphi) - w_3' \times e_v''(x, y, \varphi)\right)\right] \quad (4)$$

where, $E$ is the energy function of the fine registration; $x, y$ are the translation parameters between the DSM and OSM buildings; and $\varphi$ is the rotation parameters between the DSM and OSM buildings.

The structure of the energy function in Equation (4) is similar to the optimization function of the coarse registration. However, the fine registration considers the rotation between DSM and OSM buildings as well as more accurate translations, thus resulting in more accurate registration results. The transformation parameters $(x, y, \varphi)$, with the minimum $E(x, y, \varphi)$, are the optimal fine translation and rotation parameters for each building group. The energy function is designed to find the minimum $E(x, y, \varphi)$, since a genetic algorithm is used to find the global minimum.

2.3.2. Solution

The DSM and its gradient map are normalized for the purpose of limiting $w_1'$, $w_2'$, and $w_3'$ in the range of 0–1, and of balancing the contributions of the three scores in Equation (4). However, few extremely high buildings and noises would suppress the normalizations of the relatively lower buildings, which makes it difficult to distinguish the normalized elevations of the relatively lower buildings and the grounds. Thus, this paper suppresses those extremely high or low elevations through a series of pre-processing for a better normalization result.

This paper assumes that the ground elevations are similar in a certain local region, for example, in a region within 1 km². For a whole city, it is suggested that the city be divided into different parts according to the relief of the ground surface and that the ground elevation in each part be separately calculated. The method of calculating the ground elevation ($GE$) in a local region is introduced below.

In the pre-process step, this paper first evaluates the ground elevations from the DSM, and regards the grounds as datum to suppress those extremely high or low elevations. To achieve this goal, an elevation histogram $h_S$ with the bin interval as 3 m is defined, and $GE$ is selected from the elevation histogram $h_S$. Since the ground areas are significant when compared with the areas of other off-ground objects, the elevation label of the highest bar in $h_S$ has a high possibility to be the ground elevation. However, considering that some urban areas may have a lot of vegetation, it is possible that the vegetation area is larger than the ground area. To comprehensively select the ground elevations in different scenarios, this paper selects the highest two bars in $h_S$, and defines the ground elevation from any one of them according to the following principle: if the count of the lower bar reaches 70% of the one of the higher bar, and the corresponding elevation label of the lower bar is smaller than the one of the higher bar, $GE$ is the center value of the lower bar; otherwise, $GE$ is the center value of the higher bar. In most cases, the ground areas in cities are larger than 70% of the vegetation areas [40]; hence, the ratio between the highest two bars is decided to be 70% in the principle. The elevation histogram $h_S$ and the corresponding $GE$ are found in two experimental regions (Beijing and Shanghai), as shown in Figure 4, where the $GE$ of Beijing is 47.55 m and the $GE$ of Shanghai is 11 m. Detailed descriptions about the Beijing and Shanghai dataset are added in Section 3. To verify the validation of the $GE$ evaluations, this paper evenly selects 25 ground points in both regions, and calculates their elevation means as the true values of $GE$. The testing results show that the true $GE$ values are 46.73 m in Beijing and 10.31 m in Shanghai, respectively. The differences between the evaluated $GE$

and the true values in the two regions are both less than 1m, which will not influence the result of the experiments.

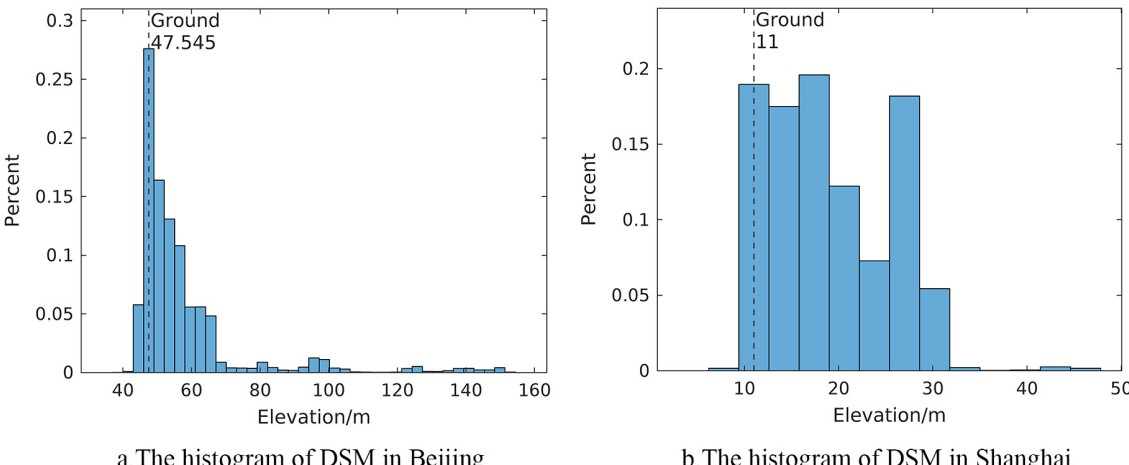

a. The histogram of DSM in Beijing      b. The histogram of DSM in Shanghai

**Figure 4.** The histogram $h_S$ in two experimental regions (Beijing and Shanghai). The ground elevation is computed by the principle mentioned above.

Given the ground elevations, the digital height model (DHM) $I_H$ is then computed by the map algebra expression $I_S - GE = I_H$, which provides the heights of off-ground objects. For a better normalization, the upper bound and the lower bound of the height ranges of $I_H$ should be limited to suppress the extremely high and low points. Due to the moving cars, repetitive textures, and so on, it is possible to find noises with extremely low heights in the DHM. To reduce the influence of these extremely low heights on the normalization, this paper sets the lower bound of the heights as $-10$ m on $I_H$, which also considers small ground height variances. The setting of the lower bound (i.e., $-10$ m) could cover the height variances of most local plain regions. For example, in a region of nearly 32 km$^2$ in Jacksonville, USA, the variance of the ground elevation is only over 7 m. To further improve the normalization accuracy, this paper refines the lower bound through the height histogram of $I_H$ with the bin interval as 1m, and removes the negative height label bins that had counts smaller than 0.01 times the maximum count in the negative height histogram.

Furthermore, high-rise buildings often refer to buildings with heights greater than 30 to 40 m [41], therefore, this paper sets the upper bound as 40 m, so that the values greater than the upper bound in $I_H$ are set as 40 m to prevent high-rise buildings affecting the normalization result. Figure 5 shows the normalization results before and after limiting the lower and upper bound, in which the result with the bounds is superior in distinguishing the grounds and buildings, with a higher lightness contrast. To quantitatively evaluate the normalization results with and without the bounding strategy, 3 points on the buildings and 3 points on the ground are selected, as shown in Figure 5, where the points with numbers 1–3 are on the buildings, and the ones with numbers 4–6 are on the ground. The values of the two normalization results are shown in a table on the right. It shows that the normalization without the bounding strategy has weak abilities to distinguish the ground and the buildings, and that some building points have similar values to the ground ones, e.g., point 3 and point 4. However, the normalization with the bounding strategy could enhance the values of the buildings and reduce the values of the ground, thus resulting in more discriminating values.

In addition, the gradient map $I_g$ is then obtained from $I_H$ using the Sobel operator, and the gradient values are limited to no more than 4 m before the normalization, since the 4 m limitation has already been able to describe the boundary gradients. After the definite upper and lower bounds, both the height map $I_H$ and the corresponding gradient

map $I_g$ are normalized, and the corresponding normalization results are defined as $I_H^n$ and $I_g^n$, respectively.

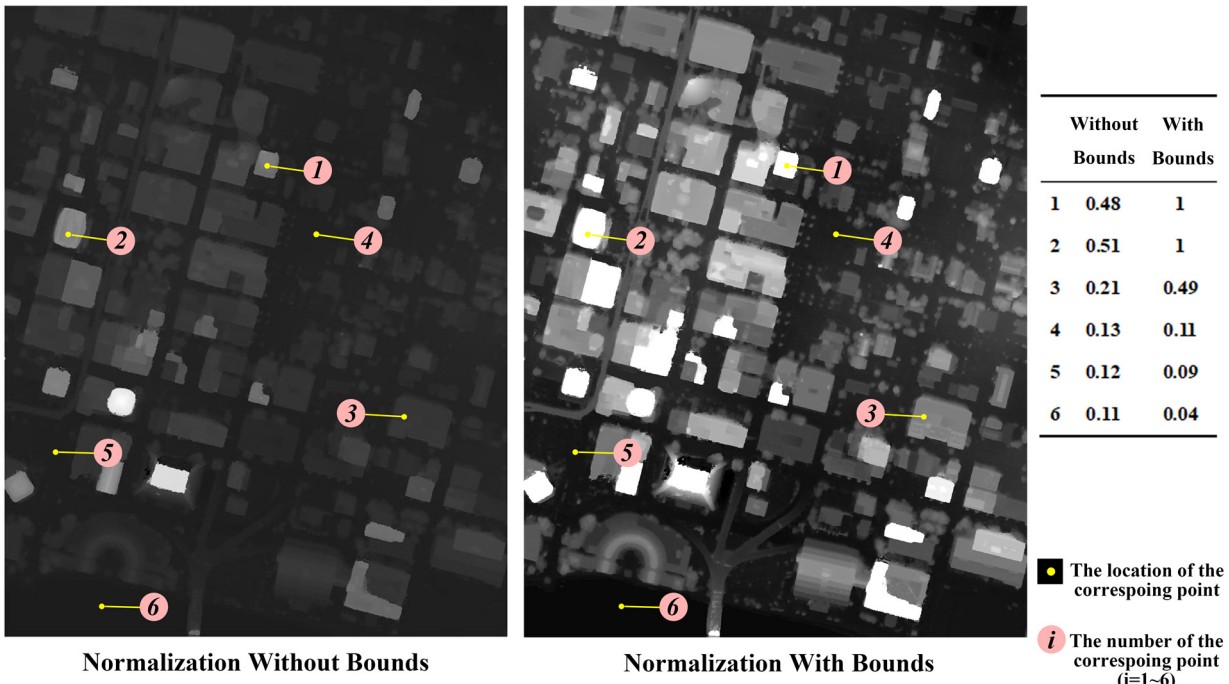

**Figure 5.** Normalization result with/without bounds in a region of Jacksonville, USA.

After the normalization, the optimal solutions of the energy function in the fine registration are computed by a genetic algorithm (GA). A continuous GA is appropriate for the optimization problem with the parameters $(x, y, \varphi)$ that have continuous and floating values, which satisfies the requirement of a high-precision solution. The GA encodes a potential solution on a simple chromosome-like data structure and applies three operators, i.e., reproduction, crossover, and mutation, to these structures, in order to preserve the critical solution information [34]. The GA is initiated by selecting a population of randomly generated solutions, then the three operators are utilized to evaluate the individuals in the population and generate successive populations that improve over time [32]. Over successive generations, the population "evolves" towards the optimal solution.

Though the GA is not dependent on the initial solutions, the good initial solution is able to limit the ranges of the population and greatly reduce the time cost of the optimization. Due to the good initial solution provided by the coarse registration, the initial population of $x$ and $y$ are randomly generated in the range of $-3s_c \leq x, y \leq 3s_c$, with $s_c$ being the moving step of the coarse registration, while the initial population of $\varphi$ is generated in the range of $-3° \leq \varphi \leq 3°$. Considering that the coarse registration results may have errors larger than the range of $-s_c \leq x, y \leq s_c$ when there are tall trees distributed around the buildings, the initial population of $x$ and $y$ in the GA is generated in the range of $-3s_c \leq x, y \leq 3s_c$, which ensures that the fine registration can correct the translation errors in the coarse registration. This paper counts the angles between OSM and the true building polygons in the two experimental regions of Beijing and Shanghai, and finds that the angle differences are mainly concentrated in the range of $-3$–$3$ degrees. Therefore, the range of $\varphi$ is defined as $-3$–$3$ degrees. The fixed maximum number of the iterations in the GA is 200, because the optimization can always be terminated within 200 iterations due to the average change in the fitness value being less than the GA function tolerance.

Considering the randomness of the initial populations in the GA, this paper runs the GA for 5 times when computing the optimal fine registration parameters for each building group, and finally takes the optimal parameters with the minimum $E(x, y, \varphi)$. The effect

of the fine registration is shown in Figure 6, where the OSM in gold is the result after the coarse registration and the OSM in pink is the result after the fine registration.

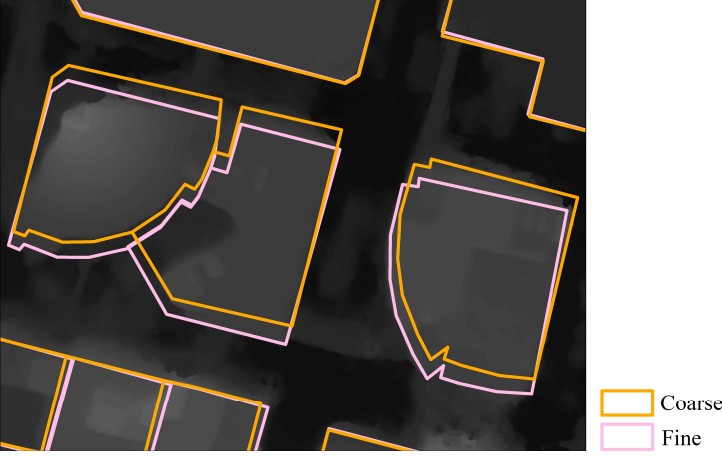

**Figure 6.** The effect of the fine registration.

*2.4. Single Building Model Reconstruction*

After the registration between OSM and the DSM, the height information from the fine registration result is obtained by clipping the DSM with the registered OSM. The rasterized building height information in the Beijing experimental region is shown in Figure 7a, where the lighter gray color refers to a higher elevation. Ultimately, the 3D building shape is generated by stretching the building height raster vertically, and then the DOM is added as texture in the generation of the single building model. The single building model in the Beijing experimental region is shown in Figure 7b.

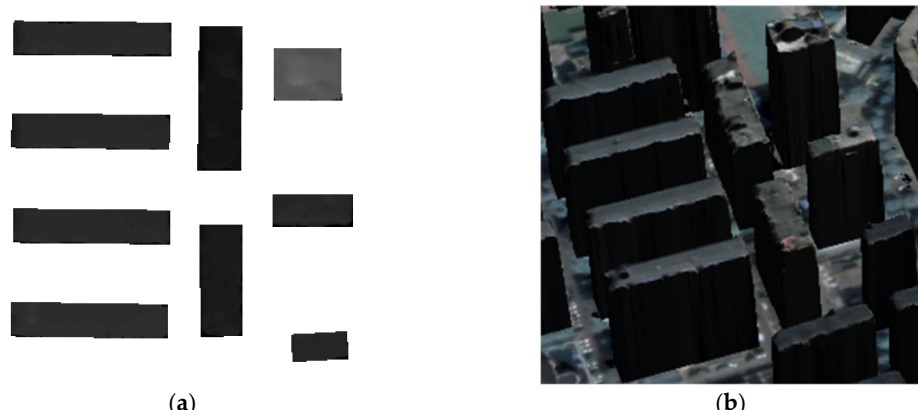

(**a**)  (**b**)

**Figure 7.** Single building model reconstruction process in Beijing experimental region, where (**a**) the building height raster is firstly generated and then (**b**) the single building model is generated.

**3. Experiments**

*3.1. Study Regions and Datasets*

In this research, the paper used the DSM datasets provided by the SSR (satellite stereo reconstruction) software [42,43] and the building polygons from the OpenStreetMap datasets, which are all converted into a UTM projected coordinate system. The study regions of these datasets include three cities: Jacksonville in the USA, and Beijing and Shanghai in China. The areas and the numbers of the buildings in each study region are shown in Table 1.

**Table 1.** The area and the number of buildings in each study region.

|  | Jacksonville | Beijing | Shanghai |
|---|---|---|---|
| Area (km$^2$) | 0.87 | 0.45 | 0.21 |
| Number of buildings | 187 | 84 | 88 |

The corresponding DSM and OSM in the three experimental areas are shown in Figure 8, where the DSMs in the three cities are generated from a WorldView-3 stereo with the GSD as 0.31 m, a GFDM-1 stereo with the GSD as 0.48 m, and a Pléiades stereo with the GSD as 0.50 m, respectively. For the Jacksonville data, the corresponding satellite stereo has been rectified by high-accuracy ground control points (GCPs) [44,45] so that there is little bias between the DSM and OSM. Thus, the original positions of OSM are regarded as true values, and various systematic registration errors (including translations and rotations) have been manually added for adjusting the parameters of the proposed method and finding the optimal ones. For the Beijing and Shanghai data, the true building polygons are manually drawn based on the corresponding digital orthophoto map (DOM), which was also derived from the SSR, as shown in Figure 8b,c. The blue polygons represent the OSM buildings and the red ones represent the true values. Due to the interim area of the building and non-building parts in the DOM, the errors of 1–2 pixels, i.e., 0.5–1 m, may occur when drawing the contours of the true building polygons. Since both the Beijing and Shanghai datasets were freely rectified without GCPs, there are various registration errors between the building polygons in OSM and the DSM when compared with the true values.

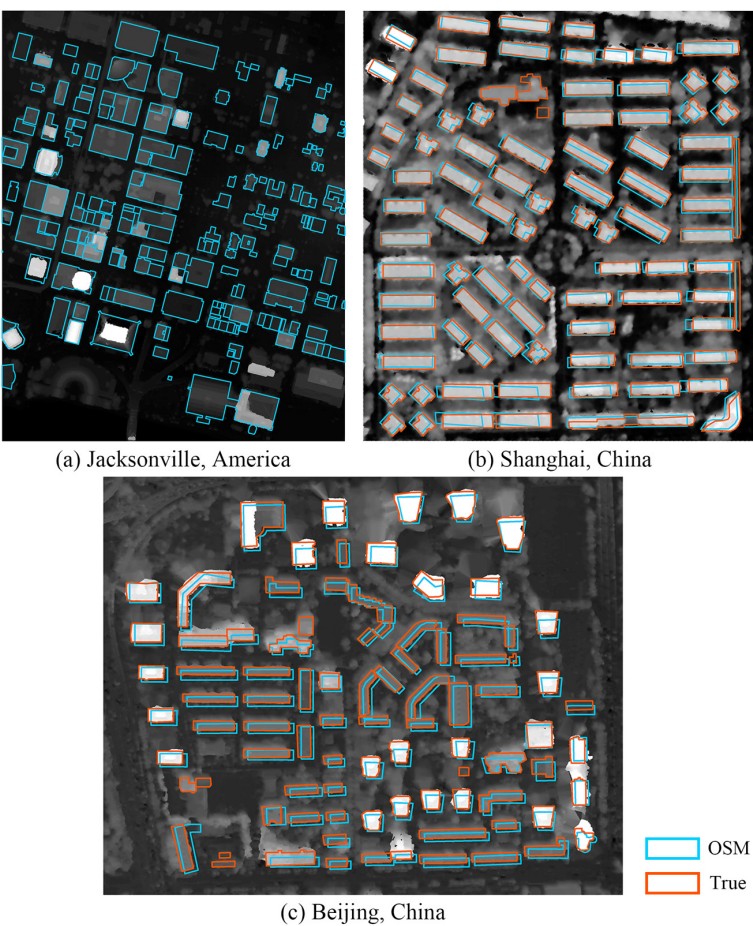

(a) Jacksonville, America

(b) Shanghai, China

(c) Beijing, China

OSM
True

**Figure 8.** The DSM, OSM, and true building polygons of the three experimental regions.

### 3.2. Accuracy Evaluation Metrics

The Jacksonville dataset was used to adjust the weight parameters in the coarse registration and fine registration, and to find the optimal ones. Since the WorldView-3 satellite imagery over Jacksonville provided by SpaceNet has already been adjusted with GCPs, the DSM from the imagery of such a high-accuracy position can fit well with the OSM. However, most commercial satellite imagery may not have available GCPs, thus the registration between the DSM and OSM is still needed. After the adjusting process, all the parameters of the proposed method were fixed, and then the experiments on the datasets of Beijing and Shanghai utilized these fixed parameters to assess the registration and reconstruction accuracy of the proposed method.

To comprehensively evaluate the performance of the proposed method, this paper utilized area-related metrics with the basic assumption that more accurate registration results should have larger overlaps between the true polygons and the registered OSM. Given the set of the true building polygons $T$ and the set of the OSM building polygons $F$ after the registration, the registration accuracy is then formulated in five aspects: (1) $IoU$, the intersection over the union, which is the intersection area of $F$ and $T$ divided by their union area; (2) precision, the intersection over the registration results, which is the intersection area of $F$ and $T$ divided by the area of $F$; (3) recall, the intersection over the true values, which is the intersection area of $F$ and $T$ divided by the area of $T$; (4) F1-score, the harmonic mean of the precision and recall; (5) $P_a$, the proportion of the buildings that have an $IoU$ larger than 0.75, which indicates the successful registration rate of the proposed method. The $IoU$, precision, recall, and F1-score were designed according to Equations (5)–(8).

$$IoU = \frac{Area(F \cap T)}{Area(F \cup T)} \tag{5}$$

$$\text{Precision} = \frac{Area(F \cap T)}{Area(F)} = \frac{TP}{TP + FP} \tag{6}$$

$$\text{Recall} = \frac{Area(F \cap T)}{Area(T)} = \frac{TP}{TP + FN} \tag{7}$$

$$\text{F1-score} = \frac{2 \times \text{Precision} \times \text{Recall}}{\text{Precision} + \text{Recall}} \tag{8}$$

where, $Area(\cdot)$ is a function to compute the area of its independent variables; $TP$, i.e., true positive, is the area of the overlap between the registration result and the true building polygons; $FP$, i.e., false positive, is the area of the parts that are within the registered OSM but beyond the true values; and $FN$, i.e., false negative, is the area of the parts that are within the true values but beyond the registered OSM.

On the other hand, in order to quantitatively evaluate the registration accuracy in a more intuitive way, this paper additionally considers the distance $\Delta_c$ between the centroids of the corresponding buildings in $T$ and $F$, as well as the angle $\Delta_\theta$ between the buildings in $T$ and $F$.

There are two common methods for calculating the angle between two polygons: (1) the MBB method: find the minimum bounding boxes (MBB) of two polygons, respectively, and calculate the angle between the long sides of the two boxes; and (2) the TPMD method: find the two boundary points with the maximum distance (TPMD) in two polygons, respectively, and calculate the angle between the two lines connecting the boundary points. However, the calculated angle will be much larger than the real one if only one method is used, especially in the cases of square-like buildings or irregular buildings. In the square-like cases, whose MBBs have a similar length of all sides, the relative long side of the corresponding polygons in the registered OSM and the true values may be different, due to some errors in OSM, the DSM, and the DOM. Thus, the MBB method may mistakenly give perpendicular angles, even though the real angle between them is close to zero, as shown in Figure 9a, where the real angle between the MBBs of the true building are in red and

the OSM after registration is in pink, and is close to $0°$, while the $\Delta_\theta$ calculated by MBB method is $89.7°$. In Figure 9a,c, the arrows are the normal vectors of the long sides in the MBBs with the same color. On the other hand, the TPMD method may fail if the shapes of the two polygons are distinct. In such a case, different boundary points with the maximum distance may be found, and $\Delta_\theta$ may be abnormally large. As shown in Figure 9d, the true building contour is distinct from OSM, and the $\Delta_\theta$ calculated by the TPMD method is $13.9°$. In most cases, the two methods are complementary, which means that the angle closest to the true value can always be found by the two methods. For example, $\Delta_\theta$ calculated by the TPMD method in Figure 9b is $0.9°$, and $\Delta_\theta$ calculated by the MMB method in Figure 9c is $0.2°$. Therefore, this paper takes the minimum angle in the two methods as the final $\Delta_\theta$.

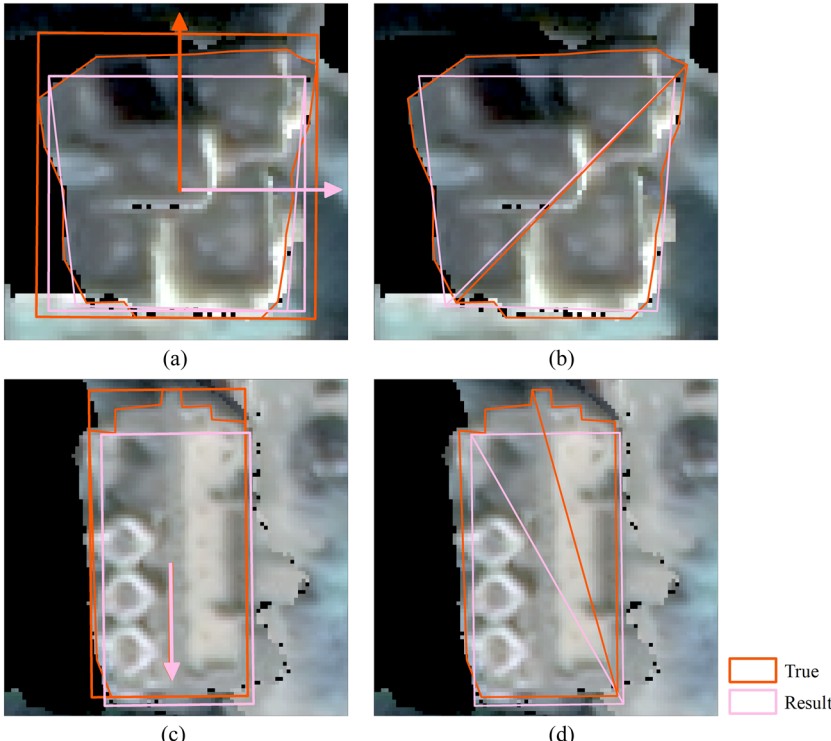

**Figure 9.** Two cases of the angle between OSM after registration and true building polygons computed by MMB method (**in the left column**) and TPMD method (**in the right column**). In (**a,c**), there are MBBs and the shapes of the true building polygons in red and the OSM after registration in pink, while the arrows are the normal vectors of the long sides in the MBBs with the same color. In (**b,d**), there are the shapes of the true building polygons and the OSM after registration, and the lines connecting the boundary points with the maximum distance.

### 3.3. Parameters Optimization

The registration performances of the coarse and the fine registrations depend on the weights of the energy functions in Equations (3) and (4). In order to find the optimal weights in the energy functions, this paper utilized the Jacksonville dataset in the weight adjustment of the coarse registration and the fine registration, respectively. Since the DSM in Jacksonville has already been registered with OSM through the use of high-accuracy GCPs, this paper first added several random translations into each group of the OSM buildings within 10 m for the optimal weight adjustment of the coarse registration, and defined these translations as true values $(\overline{d_x}, \overline{d_y})$. The proposed coarse registration method is then utilized in these translated building groups with various weights, and the weights with the registration results closest to the true values will be defined as the optimal ones. After the coarse registration, the systematic translations between OSM and the DSM are small. This paper therefore added small translations within $3s_c$ and random rotations within $-3$–$3$ degrees for the optimal weight adjustment of the fine registration, where $s_c$

is the moving step of the coarse registration. The best fine registration results are then found through comparison with the true translations $(\overline{d_x}, \overline{d_y})$ and rotations $(\overline{\varphi})$, and their corresponding weights are defined as the optimal ones.

In both the coarse and fine registrations, the optimal weights were found through grid search, which is an exhaustive search method based on the defined hyper-parameter space [46]. The grid parameters in the coarse registration and the fine registration are both the three weights in their optimization functions.

When adjusting the weights in the coarse registration, the ranges of the three weight parameters were set as $0 \leq w_1, w_2, w_3 \leq 1$, and the exploring step was set as 0.05, while the sum of the three weights was kept to 1. In total, there were 231 corresponding grids of $w_1, w_2$, and $w_3$ generated from the setting above. The different grids of the weight parameters were input into the coarse registration algorithm, and the translation error $\Delta t$ for each grid was then calculated. $\Delta t$ is the sum of the absolute differences between the translation parameters computed by the coarse registration results and their corresponding true values, as shown in Equation (9).

$$\Delta t(w_1, w_2, w_3) = \frac{\sum_{i=1}^{n} \left| d_x^i - \overline{d_x^i} \right| + \left| d_y^i - \overline{d_y^i} \right|}{n} \tag{9}$$

where, $\Delta t$ is the translation error; $w_1, w_2$, and $w_3$ are the three weights of the coarse registration; $n$ is the total number of building groups; $i$ refers to a certain building group; $d_x^i$ and $d_y^i$ are the translation parameters computed by the coarse registration; and $\overline{d_x^i}$ and $\overline{d_y^i}$ are the corresponding true translation parameters.

In a certain adjusting process, this paper calculated $\Delta t$ of the 231 groups of weights and then sorted them in ascending order to obtain the rankings of each group of weights, and finally, the group of weights with the minimum $\Delta t$ would be ranked first. Since the OSM buildings were translated randomly, the weights ranked first with the minimum $\Delta t$ might be different in each adjusting process. Therefore, this paper ran the adjusting process 10 times to get the mean ranks of each group of weights, and the group of weights with the minimum mean rank were selected as the optimal weights. The mean ranks and the mean $\Delta t$ of each group of weights in the 10 processes are shown in Figure 10, where a circle has its corresponding coordinate ($w_1, w_2$, and $w_3$) and the color of it represents the value of the corresponding parameter. Since the sum of the three parameters is kept to 1, all of the circles are located on an inclined plane in the coordinate system. If the color of a circle is bluer, its mean rank and the mean error are smaller, which corresponds to a higher registration accuracy.

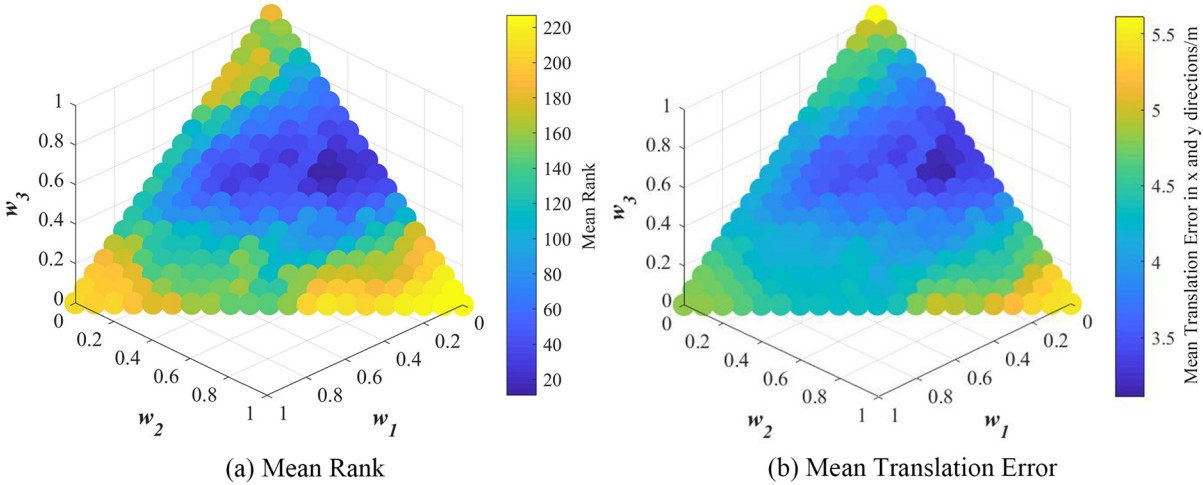

(a) Mean Rank            (b) Mean Translation Error

**Figure 10.** The registration result of all groups of weights in the coarse registration.

It is evident that Figure 10a is consistent with Figure 10b, with the circles with the low rankings in blue and the ones with the small translation errors in blue aggregated in the same place in the weight parameters space. The optimal weights in the coarse registration are: $w_1 = 0.15, w_2 = 0.4,$ and $w_3 = 0.45$, and the mean translation error of these three weights is 3.187 m.

When adjusting the weights in the fine registration, the setting of the grids was the same as in the coarse registration, i.e., there were also 231 grids of $w'_1, w'_2,$ and $w'_3$ for the weights of the fine registration. Similar to the adjusting process in the coarse registration, the different grids of the weight parameters were input into the fine registration algorithm, and the translation error $\Delta t$ and the rotation error $\Delta r$ for each grid were calculated. Here, the $d^i_x$ and $d^i_y$ in the calculation of $\Delta t$ are the translation parameters computed by the fine registration. $\Delta r$ is the sum of the absolute differences between the rotation parameters computed by the fine registration and the corresponding true values, as shown in Equation (10).

$$\Delta r\left(w'_1, w'_2, w'_3\right) = \frac{\sum_{i=1}^{n}|\varphi_i - \overline{\varphi_i}|}{n} \tag{10}$$

where, $\Delta r$ is the rotation error; $w'_1, w'_2,$ and $w'_3$ are the three weights of the fine registration; $n$ is the total number of building groups; $i$ refers to a certain building group; $\varphi_i$ is the computed rotation parameters for building group $i$; and $\overline{\varphi_i}$ is the corresponding true rotation parameter.

In a certain adjusting process, this paper calculated $\Delta t$ and $\Delta r$ of the 231 groups of weights. For the purpose of combining $\Delta t$ and $\Delta r$ to obtain the optimal weights, both $\Delta t$ and $\Delta r$ were normalized, and the corresponding normalization results were defined as $\Delta t^n$ and $\Delta r^n$, respectively. Since the satellite position error is mainly the translation error, this paper defined the larger weight of $\Delta t^n$ as 0.6 and the smaller weight of $\Delta r^n$ as 0.4, and the final error $s$ in the fine registration was defined in Equation (11). This paper sorted $s$ in ascending order to obtain the rankings of each group of weights, and the group of weights with the minimum $s$ was ranked first. Due to the randomness in the generation of the true translation and rotation parameters, this paper ran the adjusting process 10 times to get the mean ranks of each group of weights, and the group of weights with the minimum mean rank were the optimal weights. The mean rank and the mean $\Delta t$ and $\Delta r$ of each group in the processes completed 10 times are shown in Figure 11.

$$s\left(w'_1, w'_2, w'_3\right) = 0.6\Delta t^n + 0.4\Delta r^n \tag{11}$$

where, $s$ represents the error in the fine registration; and $w'_1, w'_2,$ and $w'_3$ are the three weights of the fine registration.

The same as the result of the coarse registration, the region in the darkest blue in Figure 11a is the same as Figure 11b, which demonstrates that the mean ranks are mainly decided by the translation errors. From Figure 11c, it can be seen that there are plenty of groups with a mean rotation error of less than $1°$, and that the differences among them are small. Therefore, the rotation error has little effect on the final rankings. Finally, the optimal weights of the fine registration are: $w'_1 = 0.35, w'_2 = 0.25,$ and $w'_3 = 0.4$, with their mean translation error being 2.077 m and the corresponding mean rotation error being $0.866°$.

The experimental results also show that the translation error in the fine registration is more than 1 m less than that of the coarse registration. In all the following experiments, the weight parameters in both the coarse and fine registration were fixed with $w_1 = 0.15, w_2 = 0.4,$ and $w_3 = 0.45$ in the coarse registration, and $w'_1 = 0.35, w'_2 = 0.25,$ and $w'_3 = 0.4$ in the fine registration.

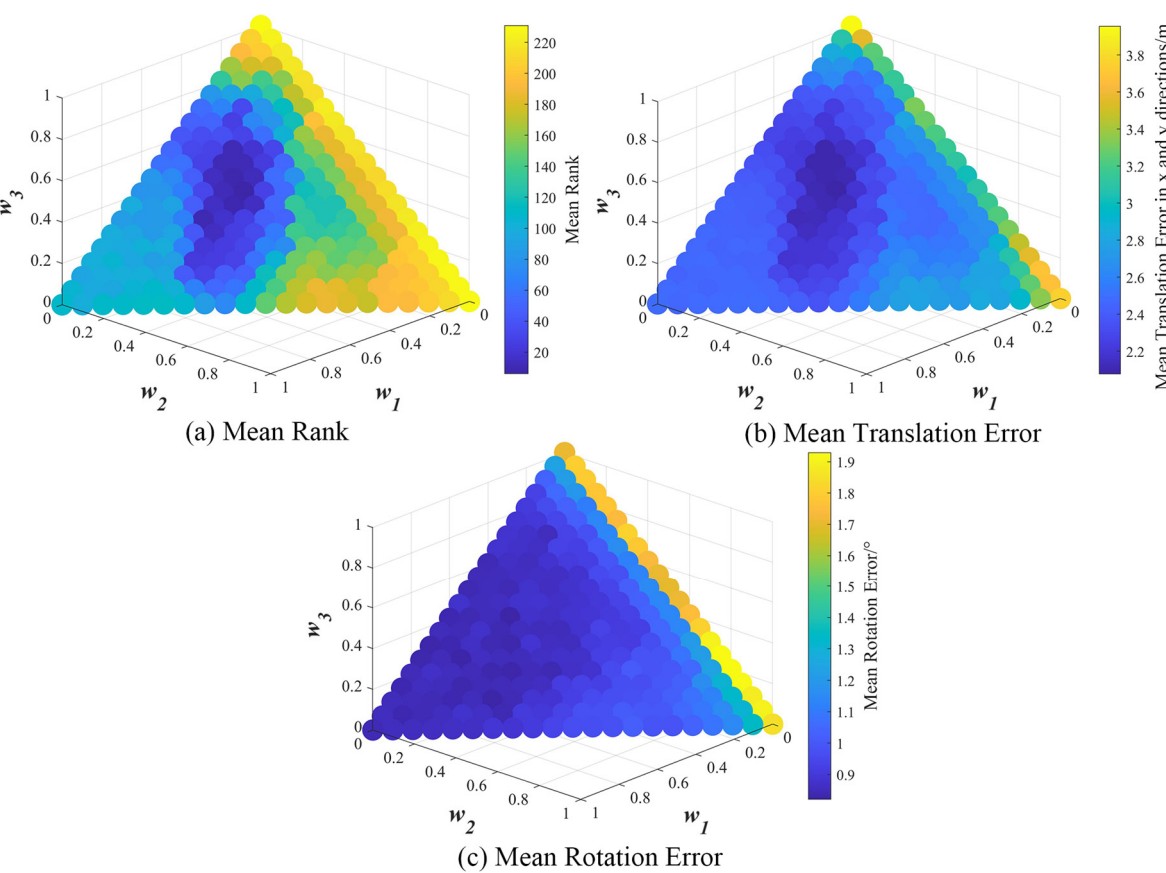

(a) Mean Rank      (b) Mean Translation Error

(c) Mean Rotation Error

**Figure 11.** The registration result of all groups of weights in the fine registration.

### 3.4. Experimental Result Analysis

In order to comprehensively test the performance of the proposed method in single building reconstruction, the registration results in the Beijing and Shanghai datasets were evaluated qualitatively and quantitatively. In general, this paper compared the registration results with the true values, and evaluated the results within the metrics of *IoU*, precision, recall, F1-Score, $P_a$, $\Delta_\theta$, and $\Delta_c$.

For the qualitative evaluation, Figure 12 directly shows the differences of OSM before and after the registration, where the original OSM is in blue, the registration result of the proposed method is in green, and the true building polygons are in red. The pictures on the left are the original OSM and the true values on the DOM, while pictures on the right are the registration results and true values on the DOM. According to Figure 12a,b, it is obvious that, in both regions, the original OSM has various systematic errors compared with the true value, while the registration result basically coincides with the true value. The transformation of OSM after the registration indicates that our proposed method has brought OSM closer to the true building polygons, which offers a high-quality reconstruction.

For the quantitative evaluation, since the OSM and true building polygons were stored as a shapefile format, all the metrics mentioned in Section 3.2, except for $P_a$, are first computed for each single building, and then the accuracy metrics of all the buildings are averaged to evaluate the registration results. The testing results with the different metrics in the Beijing and Shanghai regions can be found in Table 2, which demonstrates that our proposed method produces a high-precision single building model. Among the area-related metrics in the two regions, after the registration of the proposed method, the *IoU* was increased by 69.8%/26.2%, the precision was increased by 41.0%/15.5%, the recall was increased by 41.0%/16.0%, and the F1-score was increased by 42.7%/15.8%, compared with the situation of using the original OSM before the registration, which indicates that the proposed method effectively improves the overlap rate between the OSM and true values

from different aspects. Since the registration error in the Shanghai region is smaller than the one in the Beijing region, i.e., the accuracy of the original OSM is higher, the percentage of improvement is lower than the one in Beijing.

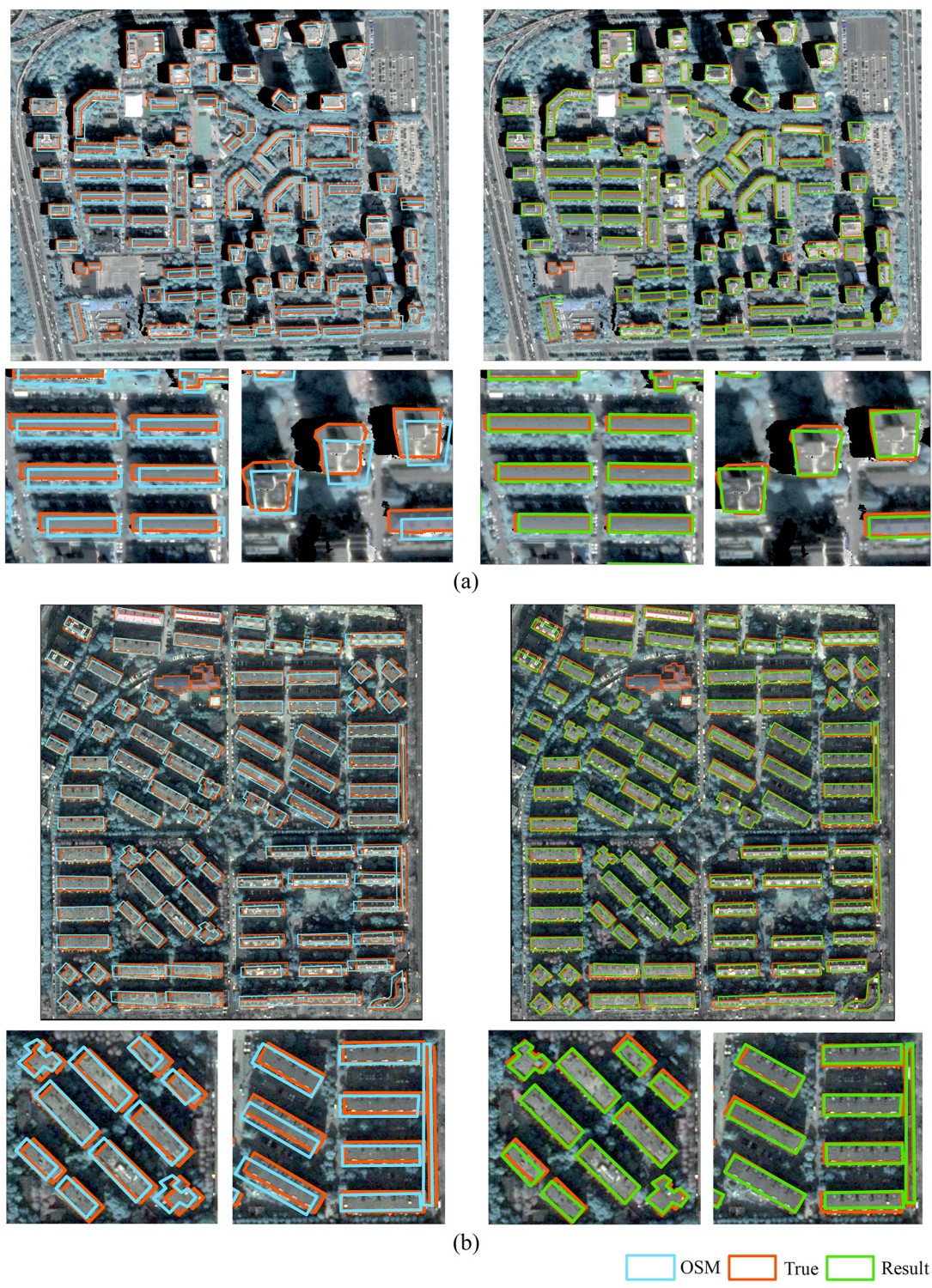

**Figure 12.** The OSM before and after registration compared with true building polygons in (**a**) Beijing, and (**b**) Shanghai experimental regions. In both regions, two sub-areas are enlarged and placed below the full map to better show the registration result.

**Table 2.** Accuracy metrics before and after registration in Beijing and Shanghai regions.

| Region | | *IoU* | Precision | Recall | F1-Score | $P_a$ | $\Delta_C$/m | $\Delta_\theta$/° |
|---|---|---|---|---|---|---|---|---|
| Beijing | Before | 0.454 | 0.627 | 0.605 | 0.613 | 0.012 | 6.900 | 1.654 |
| | After | 0.771 | 0.884 | 0.853 | 0.875 | 0.655 | 2.244 | 1.116 |
| Shanghai | Before | 0.618 | 0.794 | 0.724 | 0.754 | 0.148 | 4.388 | 1.340 |
| | After | 0.780 | 0.917 | 0.840 | 0.873 | 0.659 | 1.573 | 1.112 |

$P_a$, the percentage of the buildings with an *IoU* higher than 0.75 in the region, directly shows the proportion of the buildings with a high overlap rate. After the registration, $P_a$ increased from 0.012 to 0.655 in the Beijing region, and from 0.148 to 0.659 in the Shanghai region, which indicates that, in both regions, more than half of the buildings were not registered well before being accurately overlapped with the true values via the registration.

In addition, the registration accuracy can be evaluated intuitively from the metrics $\Delta_C$ and $\Delta_\theta$. The $\Delta_C$ of OSM before the registration in the Beijing/Shanghai regions were 6.900 m /4.388 m, and they were decreased by 4.656 m/2.815 m through the registration of the proposed method, which demonstrates that the proposed method can successfully reduce the translation error in OSM. After the registration, the $\Delta_C$ in Shanghai was 1.573 m, which is smaller than the mean translation error in the fine registration (2.077 m) in Section 3.3, while the $\Delta_C$ in Beijing was 2.244 m, a little bit larger than 2.077 m. $\Delta_C$ is not only determined by the registration result, but also by whether the shapes of OSM and the true building polygons are similar. If the shapes of them are different, their centroids are generally in different locations and the calculated $\Delta_C$ will be large. There are more buildings that have different shapes between OSM and the true building polygons in the Beijing region, as shown in Figure 12, which is the reason why the average $\Delta_C$ after the registration in Beijing is larger than that in Shanghai. In fact, the reduction of the $\Delta_C$ in the Beijing region (4.656 m) is much larger than the one in the Shanghai region (2.815 m).

The $\Delta_\theta$, after the registration in the Beijing and Shanghai regions, were 1.116° and 1.112°, which were decreased by 0.538° and 0.228°, respectively, compared with the $\Delta_\theta$ of the original OSM, which were 1.654° and 1.340°. Though the improvement of $\Delta_\theta$ is not as substantial as the area-related metrics, it still indicates that the proposed method corrects the rotation error in OSM to a certain extent. In addition, the differences between the shapes of OSM and the true building polygons may cause abnormally large angles in some cases, which is mentioned in the introduction to the two methods of calculating $\Delta_\theta$; hence, the practical $\Delta_\theta$ should be smaller than the calculated value. Even though this paper takes the minimum angle of the two methods, there are still a few cases where the $\Delta_\theta$ is larger than the real angle. Figure 13 shows some buildings in the Beijing region having irregular shapes, with the $\Delta_\theta$ larger than 3°.

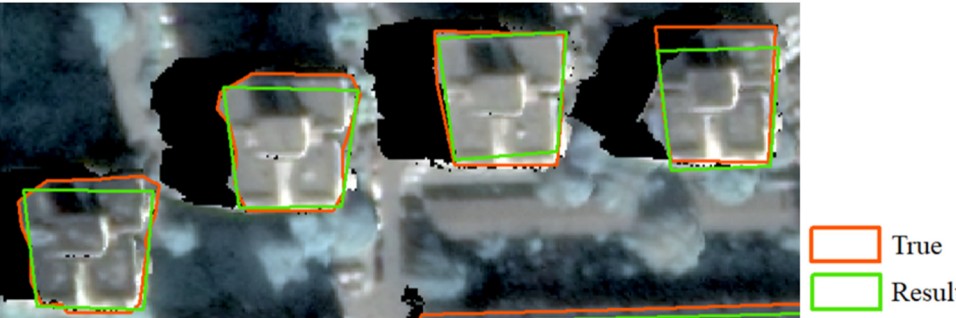

**Figure 13.** The buildings with $\Delta_\theta > 3°$ in Beijing region.

3D single building models in the Beijing and Shanghai regions are visualized in Figure 14, which indicates that our proposed method can reconstruct single building models with high-accuracy contours.

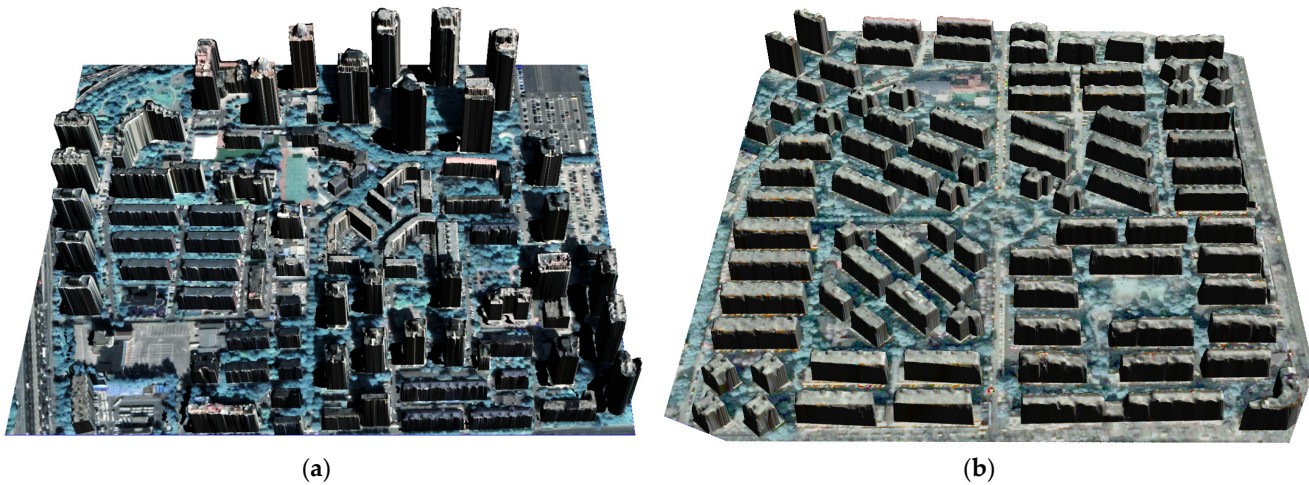

<div align="center">(<b>a</b>)                             (<b>b</b>)</div>

**Figure 14.** 3D single building model in (**a**) Beijing, and (**b**) Shanghai region.

With regards to the aspect of computation efficiency, the program run time in the three experimental regions on MATLAB, without using parallel pool, was calculated on a device with a CPU of AMD Ryzen 9 7950X 16-Core Processor. In the Jacksonville region, with area of 0.87 km$^2$, the program ran for 57 s; in the Beijing region, with area of 0.45 km$^2$, the program ran for 33 s; and in the Shanghai region, with area of 0.21 km$^2$, the program ran for 36 s. It is evident that the proposed method has a low time-cost in a single-core process. At present, the proposed method is realized on a single-core program. In the future, this study intends to improve the algorithm to a multi-core parallel framework, which is expected to further lower the time cost.

*3.5. Comparison*

In this section, the paper compared the OSM registration results obtained from the proposed method with the building extraction results from several convolutional neural networks (CNN) using satellite imagery, including HRNet + OCR + SegFix [47], Deeplab v3+ [48], and UNet [49]. Since the results from the CNNs were in raster format, this paper rasterized the registration results in the Beijing and Shanghai experimental regions for comparison, and the pixel-based metrics of *IoU*, precision, recall, and the F1-score were utilized in the accuracy comparisons, where *T* refers to all the pixels in the true values and *F* refers to all the pixels in the results of the different methods.

Table 3 demonstrates the significant advantage of our proposed method in single building reconstruction when compared with the traditional CNN methods. In both of the experimental regions with different building distributions, our proposed method had the best *IoU*, which were 0.775 and 0.774, respectively, while the performances of the CNN were obviously inferior and unstable, with the *IoU* values all lower than 0.6 and differences in the *IoU* values evident in the two regions. For UNet, the difference reached 0.228, indicating that it performed unevenly in a semantic segmentation of the two regions; for HRNet + OCR + SegFix/Deeplab v3+, the difference was 0.131/0.142, reflecting the networks' deficiencies in their generalization ability. Among all the metrics, our proposed method achieved the highest score for the *IoU*, precision, and F1-score, but had the lowest score for recall. Recall is the intersection area of *F* and *T* divided by the area of *T*, and this is proportional to the intersection area due to the area of *T* being fixed. The rasterized result of this paper, as well as the building extraction results of different CNNs, are visualized in Figure 15, where the extraction results of the CNN contained a lot of non-building areas. Since the CNN results included the true building area and other non-building areas, the intersection area of *F* and *T* should be closer to the area of *T*, which raised their corresponding values of recall.

**Table 3.** Pixel-based evaluation on our proposed method and different CNNs.

| Region | Metric | Ours | HRNet + OCR + SegFix | Deeplab v3+ | UNet |
|---|---|---|---|---|---|
| Beijing | *IoU* | 0.775 | 0.331 | 0.457 | 0.290 |
| | Precision | 0.895 | 0.335 | 0.495 | 0.295 |
| | Recall | 0.852 | 0.964 | 0.857 | 0.948 |
| | F1-score | 0.873 | 0.498 | 0.628 | 0.450 |
| Shanghai | *IoU* | 0.774 | 0.462 | 0.599 | 0.518 |
| | Precision | 0.924 | 0.476 | 0.672 | 0.561 |
| | Recall | 0.827 | 0.943 | 0.847 | 0.870 |
| | F1-score | 0.873 | 0.633 | 0.749 | 0.682 |

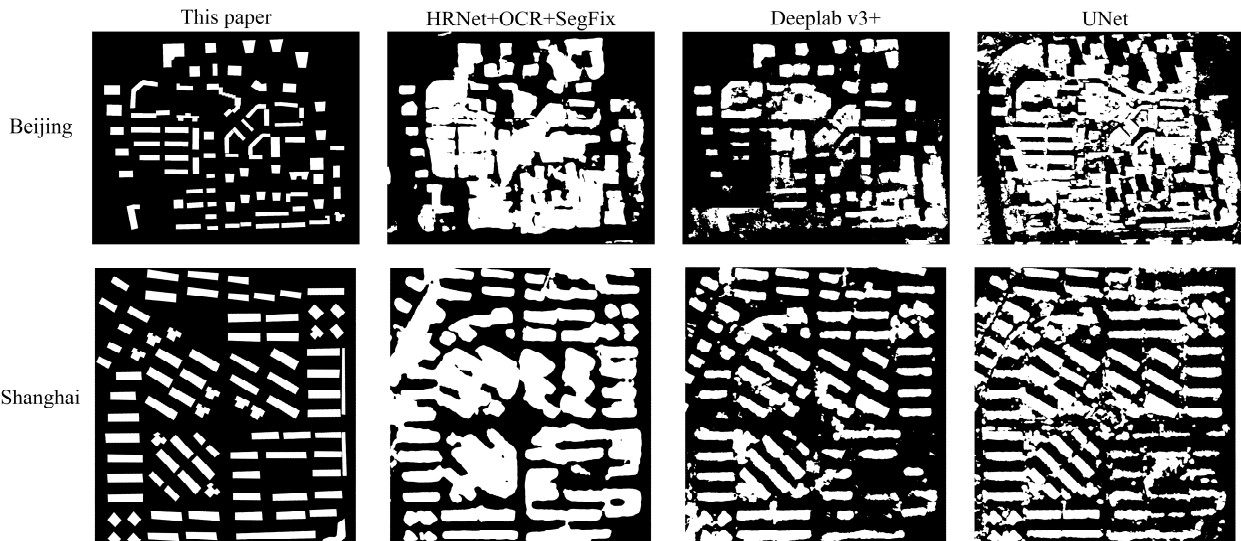

**Figure 15.** The comparison of results in this paper and different convolutional neural networks.

Additionally, Figure 15 indicates that the building model produced by the proposed method in this paper had relatively clear building contours, as well as less noises, while the CNN results contained non-building areas caused by misclassification, which is useless for building reconstruction. In general, our proposed method outperformed the traditional CNN works in the qualitative and quantitative evaluations.

## 4. Discussion

The experimental results of this study indicate that the proposed method can greatly reduce the systematic error between OSM and the DSM in regions with different building types and distributions. The OSM results before and after the registration in the Beijing and Shanghai regions are shown in Figure 12, which indicates that OSM after the registration basically coincides with the true values. The quantitative evaluation of both regions is shown in Table 2. Compared with the original OSM, the *IoU* was increased by 69.8%/26.2%, the precision was increased by 41.0%/15.5%, the recall was increased by 41.0%/16.0%, and the F1-score was increased by 42.7%/15.8%, which demonstrates that the proposed method effectively improves the overlap rate between OSM and the true values from different aspects. Additionally, the $\Delta_C$ and $\Delta_\theta$ in both regions were decreased by 4.656 m/2.815 m and 0.538°/0.228°, respectively, indicating that the proposed method successfully reduced the translation error and the rotation error in OSM. The experimental comparisons were also tested on the datasets of the Beijing and Shanghai regions. Compared with the building segmentation results from the CNNs mentioned in Section 3.5, the proposed method achieved the highest *IoU*, precision, and F1-score, which indicates that the building footprints produced by the proposed method had the highest overlap rate with the true

values. In addition to the overlap rate, the single building model in the proposed method also had the most accurate building contours, since OSM provides high-precision and complete building polygons, which is shown in Figure 15, where the CNN results had unclear contours and non-building areas.

This study innovatively proposed a two-step "coarse-to-fine registration" algorithm to reduce the systematic error of OSM, and can effectively produce a high-precision single building model with OSM after the registration and DSM. The proposed method can produce large-scale building models globally, and has a great potential to be applied in some urban and 3D applications, e.g., urban planning, environmental monitoring, virtual city tours, and national defense military.

The proposed method has some limitations to be considered. First, there are several manually selected parameters in the fine registration, which can be further simplified. For example, in the ground elevation selection and bounding strategy of the DHM before the normalization in the fine registration, some parameters are manually set. We plan to optimize the algorithm with fewer manually set parameters in the future. Second, the study has not solved the problem of shape distortion. A DIM-derived DSM often meets edge-fattening issues, that is, the building shape in the DSM has slight differences compared to that in OSM. If the shape distortion in this case is considered, the registration and reconstruction accuracy may decrease, since the OSM is adjusted based on the DSM in the proposed method. Considering this existing issue, we plan to use deep learning technology that combines a DOM and OSM to predict high-precision building contours in future work. Third, the registration accuracy of the proposed method may be relatively low in areas with many trees. Man-made structures (e.g., buildings) can generally represent good persistent scatterers [50,51], which are distinguished to vegetation. In future work, we plan to utilize the different scattering characteristics of buildings and other objects such as vegetation to improve the registration and modeling accuracy.

## 5. Conclusions

This paper proposed a novel registration method between OSM and the DSM from satellite stereos by formulating the two-step "coarse-to-fine registration" into the optimization of their energy functions. The main contribution of the proposed method is to address the mis-registrations between the DSM and OSM and innovatively propose a low time-cost, scalable, and high-accuracy single building model reconstruction method based on the DSM and OSM. Experiments on Beijing/Shanghai datasets show that the proposed method significantly improves the overlap rate of the registration results and the true values, and reduces the OSM translation error by 4.656 m/2.815 m, and the rotation error by 0.538°/0.228°. When compared with CNN building segmentation methods, the proposed method could achieve the highest overlap rate with the true values, and generate a single building model with the most accurate building contour. However, the reconstruction accuracy of the proposed method is influenced by whether the shapes of OSM are consistent with the DSM. In future work, the OSM vertices movement method will be developed.

**Author Contributions:** Conceptualization, Y.H. and X.H.; methodology, Y.H. and X.H.; software, Y.H.; validation, Y.H., W.L. and X.H.; formal analysis, Y.H.; investigation, Y.H.; resources, X.H.; data curation, X.H.; writing—original draft preparation, Y.H.; writing—review and editing, Y.H., X.H. and H.H.; visualization, Y.H. and W.L.; supervision, X.H.; project administration, X.H.; funding acquisition, X.H. All authors have read and agreed to the published version of the manuscript.

**Funding:** This research was funded by the National Natural Science Foundation of China (grant number 41701540), Basic Startup Funding of Sun Yat-sen University (grant number 76230-18841205), and 3D reconstruction technology based on high-resolution aerospace line array images (grant number 76230-71020009).

**Data Availability Statement:** Some of the data presented in this study are openly available. The OSM data is available in https://www.openstreetmap.org, and the WorldView-3 data is available

in https://spacenet.ai/core3d/. Due to the requirements of the data provider, the sharing of the Pleiades stereo and the GFDM stereo is not applicable to this article.

**Acknowledgments:** The authors would like to thank the intelligence advanced research projects activity (IARPA) and DigitalGlobe for providing the WorldView-3 satellite imagery in the CORE3D public dataset.

**Conflicts of Interest:** The authors declare no conflict of interest.

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
