# Peer review of "High-Precision Single Building Model Reconstruction Based on the Registration between OSM and DSM from Satellite Stereos"

_remotesensing, doi:10.3390/rs15051443_

Round 1

Reviewer 1 Report

This manuscript is generally well prepared, while there are a few issues need to be further clarified. Please see below:

1.          The article has emphasized single building reconstruction while only the contour of each individual building within certain areas were reconstructed. Normally reconstruction means the detail rendering of the object’s geometry/façade rather than the contour.

2.          Page 4 line 167: how do one determine how many times of GSD to be used here?

3.          Page 5 Figure 2(a): how the boundary points/polygons (straight, closed polygons) were obtained?

4.          Page 6: it is kind of confusing that if the weighting w1, w2, w3 were solely to account for the building area effects in the index calculations, should they not be the same for one building? i.e. w1 = w2 = w3

5.          Page 7 line 268-273 and Page 8 Figure 4(b): the figure actually shows multiple peaks – if not known in prior, how can one assure 11 m is the GE of Shanghai?

6.          Page 9 line 328-331: it is confusing that why in fine registration the range of x y in GA are 3 times (even larger) the moving step in the coarse one.

7.          Page 10 Figure 6: should the legend “Rough” be “Coarse” instead?

8.          Figure 9 and 10: it is kind of difficult for readers to understand what they really represent. A 2D color contour in a 3D coordinate?

9.          Page 18 line 583-587 and Figure 13: are these results directly derived from the author’s methods? It is difficult for readers to see the correlation.

Reviewer 2 Report

The manuscript presents a new method for optimizing the registration between rasterized DSM and building footprints in OSM.

The manuscript is well thought out, the references are complete and up to date, and the methodology is well described.

The results of a three-zone test show good performance.

Just an observation. The authors assume that surrounding buildings with short spatial distances have similar registration errors and assign the same registration parameters (for both coarse and fine registrations) to polygons within the same group during the registration process. There are often other constraints, due (e.g.) to the alignment of buildings along the same street. Why didn't you consider also this type of constraint to group buildings?

Reviewer 3 Report

The paper proposes an automated procedure for reconstructing the 3D shapes of buildings (including footprint and height) through digital surface maps obtained through high-resolution satellite images and building footprints in OpenStreetMap. The problem is subdivided in two steps. A first coarse registration and a fine-tuning step.

Producing a high precision single building model is a very useful purpose which may be synergistic with other remote sensing tools. Particularly, a major issue existing in assessing structural movements through InSAR analysis is accurately locating the persistent scatterers over the structural surface avoiding to track the movement of different objects. Since the authors mention SAR images in their introduction, it would be desirable to enlarge the discussion and mention this synergy with InSAR in their literature review. Following are some suggested studies where similar problems are discussed (e.g. for buildings and bridges): https://doi.org/10.3390/rs10071137 , https://doi.org/10.1016/j.isprsjprs.2015.10.011 and https://doi.org/10.1016/j.rse.2019.111453

The authors mention that their approach is “low-cost”. What does low-cost mean in terms of computational burden? Can the authors provide some comments on this aspect?

In the fine tuning step, rotation and translation are optimally adjusted. However, no shape distortion is introduced. This seems to be a limitation of the study. Please comment.

Please clearly state if other literature papers proposed the combination of DSM and OSM or if this is the first time that this is done.

Indicate the feasible range in the maximum operator also on the left hand side of Equation 3. Also, parentheses are missing on the right hand side. Same for other similar equations.

“Since most cities are located in plains”: please rephrase, this is not true especially in Europe.

Use relative count in figure 4.

Precision and recall values in Table 4 are pretty high but still not really close to 1.0. Is this an acceptable level of approximation for most applications? Please comment.

Avoid using acronyms in abstract and define them the first time they are used.

Round 2

Reviewer 3 Report

The authors have addressed all comments of this referee in a satisfactory way